# Sustainable Nutrition and Food Allergy: A State-of-the-Art Review

**DOI:** 10.3390/nu17152448

**Published:** 2025-07-27

**Authors:** Caterina Anania, Barbara Cuomo, Enza D’Auria, Fabio Decimo, Giuliana Giannì, Giovanni Cosimo Indirli, Enrica Manca, Filippo Mondì, Erica Pendezza, Marco Ugo Andrea Sartorio, Mauro Calvani

**Affiliations:** 1Department of Maternal Infantile and Urological Science, Sapienza University of Rome, 00161 Rome, Italy; filippo.mondi94@gmail.com; 2Central Operative Unit of Pediatrics and Allergy Center for Children and Adults, Santa Rosa Hospital, 01010 Viterbo, Italy; cuomoba@gmail.com; 3Allergy Unit, Department of Pediatrics-Buzzi Children’s Hospital, 20154 Milan, Italy; enza.dauria@unimi.it; 4Department of Biomedical and Clinical Sciences, University of Milan, 20154 Milan, Italy; 5Department of Woman, Child and General and Specialized Surgery, University of Campania Luigi Vanvitelli, 80138 Naples, Italy; fabio.decimo@unicampania.it; 6Pediatrics Department, Allergology and Immunology, Italian Society of Pediatric Allergology and Immunology (SIAIP) Regional Referent for Emilia-Romagna, 57124 Livorno Ospedali Riuniti, Italy; giannigiuliana90@gmail.com; 7Pediatric Independent Researcher, Italian Society of Paediatric Allergology and Immunology (SIAIP) Regional Referent for Puglia and Basilicata, 73100 Lecce, Italy; gindirli@libero.it; 8Pediatrics Department, Policlinico Riuniti, University Hospital of Foggia, 71122 Foggia, Italy; enricamanca@hotmail.com; 9IDESP Institute, INSERM Institute, University of Montpellier, 34090 Montpellier, France; 10Department of Pediatrics Allergy Center, IRCCS Institute Giannina Gaslini, 16147 Genoa, Italy; erica.pendezza@asst-fbf-sacco.it; 11Pediatric Allergology Unit, Department of Childhood and Developmental Medicine, Fatebenefratelli-Sacco Hospital, 20121 Milan, Italy; marco.sartorio@asst-fbf-sacco.it; 12Operative Unit of Pediatrics, S. Camillo-Forlanini Hospital, 00152 Rome, Italy; maurocalvani58@gmail.com

**Keywords:** novel food, plant-based proteins, cultured meat, single-cell proteins, edible insects, seaweed, allergenicity

## Abstract

Alternative proteins denote non-traditional, high-protein foods. These innovative sources aim to compete with conventional animal products by providing protein-rich, sustainable, nutritious, and flavorful options. Currently, five main categories of alternative proteins are being developed: plant-based proteins, cultured meat, single-cell proteins, edible insects, and seaweed. Nonetheless, several chemical and microbiological food safety hazards are associated with these alternatives Incorporating novel protein sources into food products may heighten the prevalence of existing food allergies. This could arise from extracting proteins from their natural matrices and utilizing them at significantly higher concentrations. Additionally, the introduction of new proteins may lead to the development of novel food allergies. Proteins that are currently seldom or never consumed may cause primary sensitisation or trigger cross-reactivity with known allergens. To date, alternative proteins have not been thoroughly studied for their allergenic potential, and there is no standardised method for assessing this risk. This review aims to explore non-traditional protein sources, discussing their nutritional and functional properties, as well as their potential allergenicity based on available research. We conducted a literature search in PubMed and Embase databases. We used specific keywords and MESH terms. A total of 157 studies were included in the review. The studies reviewed in our analysis reveal significant limitations, such as inconsistent methodologies, limited participant numbers, and a lack of long-term data, which hinder the ability to make clear conclusions regarding the safety of these new proteins for individuals with allergies. To address current challenge, future research should integrate food science, regulatory perspectives and advanced technologies.

## 1. Introduction

The global population is expected to reach 10 billion people by 2050. Therefore, one of the most pressing problems for humanity is how to feed the world population while preserving the planet [1]. The rapid population growth has raised concerns about whether the Earth’s finite resources, such as agricultural land and freshwater, will be sufficient to satisfy the food demands of such a large population [2]. In addition, the rising consumption of meat and fish is considered problematic, primarily because intensive farming practices are often associated with public health risks, sustainability challenges, and environmental concerns (including greenhouse gas emissions, land and water consumption), as well as animal welfare issues (such as the impact of fertilizers, pesticides, and hormones used in animal feeding) [3]. Numerous studies suggest that many of these critical issues could be substantially alleviated by removing animals from the production process. Consequently, considerable efforts are being devoted to identifying alternative protein sources, essential for human nutrition. These alternative proteins encompass a range of options, including edible insects, plant-based proteins, fungal proteins, algae proteins, cultured meat and single-cell proteins. In recent years, investment in the alternative protein sector has rapidly increased, with hundreds of new companies entering the market and billions of dollars being invested. A recent report indicates that the alternative protein market is expected to reach US$ 290 billion by 2035, accounting for 11% of the overall protein market [4]. However, it is essential to carefully assess and identify potential risks to consumer health when incorporating alternative proteins into the food system. One of the primary concerns is the potential for allergens to enter the diet through the consumption of novel or modified proteins or through allergen cross-reactivity (Figure 1). Food allergy (FA) is an increasing health concern globally. At present, 10% of adults and 8% of children in developed countries are affected by this condition. FA is an immune system reaction that occurs in sensitized individuals whenever they are exposed to a specific allergen. It is distinct from food intolerance, which involves a non-immune response. FA can be classified into three types: IgE-mediated, caused by immunoglobulin E (IgE) antibodies targeting food antigens; non-IgE mediated, where the immune response primarily involves cell-mediated mechanisms; and mixed, where both IgE-mediated and cell-mediated immune responses are involved [5]. IgE-mediated FA are the most common. They are easily characterized by the presence of specific serum IgE (sIgE) or a positive skin prick test (SPT). They occur most frequently in the first years of life, giving rise to urticaria/angioedema, oral allergic syndrome, rhinitis, or acute asthma and anaphylaxis [6]. Non-IgE FA are characterised by cutaneous reactions (such as atopic dermatitis, contact dermatitis and dermatitis herpetiformis), respiratory reactions (such as Heiner’s syndrome) or gastrointestinal reactions [7]. Non-IgE and mixed FA have a less clear pathogenesis, despite their frequency. The clinical manifestations of food allergies include skin, gastrointestinal, respiratory and systemic manifestations with varying severity [8,9,10]. The diagnostic approach begins with a careful medical history collection to identify the culprit food and a physical examination [11]. Oral food challenge (OFC) currently represents the diagnostic gold standard, even if the accuracy of diagnostic tests for some foods, when combined with clinical history, may be sufficient in many cases in clinical practice to provide a robust diagnosis [12,13]. Treatment consists of the elimination of the offending allergen. The development of allergic diseases depends not only on the allergen itself but also on the individual genetic predisposition and other environmental factors. Currently, no validated tests are available to predict the sensitization potential of food proteins [14]. The current available methodology is suitable mainly for assessing new proteins’ allergenic potential for cross-reactivity. In contrast, methodologies to evaluate the allergenic potential of new proteins due to de novo sensitisation are limited [15]. Most available tests focus on sequence similarity, which may elucidate a cross-reactive potency but not de novo sensitization [16]. Current valuation strategies for the allergenicity of alternative proteins primarily depend on a weight-of-evidence approach to risk assessment [17,18,19,20]. According to EFSA, it includes assessing the gene sources, analyzing sequence homology with known allergens, testing the binding affinity to IgE from allergic individuals, and evaluating the protein stability using a pepsin resistance test. Furthermore, the European Food Safety Authority (EFSA) recommended that allergenicity assessments consider the production process and available experimental and human data, including cross-reactivity information. Risk assessment of de novo sensitization presents a greater challenge and necessitates the development of reliable and validated in silico, in vitro and in vivo models (Figure 1). This review aims to provide a comprehensive overview of the most promising alternative proteins sources, highlighting the possible risks in term of their potential allergenicity and critically discussing potential clinical implications.

## 2. Methodology

A comprehensive literature search was conducted in April 2025 using the MEDLINE database via PubMed (www.pubmed.gov accessed on 26 April 2025) and Embase (www.embase.com accessed on 27 April 2025).

No restrictions were applied regarding language, publication type, study design, or year of publication. Additional relevant articles were identified by manually screening the reference lists of the included studies. Guidelines, systematic reviews, and original research articles were all considered for inclusion. We employed a standardized search strategy across all categories of alternative proteins, modifying only the protein-specific keywords while maintaining a consistent focus on allergenicity.

The following search strings were used:

Insects: (“insect protein” OR “edible insects”) AND (“allergenicity” OR “allergic reactions” OR “food allergy”); Seaweeds: (“seaweeds” OR “algae” OR “sea vegetables”) AND (“allergenicity” OR “allergic reactions” OR “food allergy”); Cultured meat: (“cultured meat” OR “lab-grown meat” OR “cell-based meat” OR “in vitro meat”) AND (“allergenicity” OR “allergic reactions” OR “food allergy”); Single-cell protein: (“single cell protein” OR “SCP” OR “microbial protein” OR “yeast protein” OR “fungal protein”) AND (“allergenicity” OR “allergic reactions” OR “food allergy”); Plant-based protein: (“plant-based protein” OR “plant protein” OR “vegetable protein”) AND (“allergenicity” OR “allergic reactions” OR “food allergy”). The search yielded 292 articles from PubMed and 405 from Embase. After removing duplicates, a total of 315 unique records were identified. A total of 157 studies were included in the Review (Figure 2).

## 3. Results and Discussion

### 3.1. Cutured Meat

An emerging alternative to conventional animal products, cultured meat, also known as cultivated, synthetic, or in vitro meat, is increasingly recognized as a viable alternative to traditional animal-derived products [21]. This innovative technology has garnered significant interest over the past decade due to its potential to mimic the nutritional value and sensory characteristics of conventional meat while addressing the environmental and ethical issues associated with intensive livestock farming. Cultured meat production avoids many of the greenhouse gas emissions, water use, and land degradation caused by traditional farming methods. Moreover, it offers a solution to the moral dilemmas tied to the exploitation and slaughter of animals, presenting an alternative that could transform the food industry. The production process begins with the isolation of stem cells from a small biopsy taken from a live animal [22]. These cells are then cultivated in a nutrient-rich medium under laboratory conditions, where they multiply and differentiate into muscle or fat cells [23]. By growing the cells on three-dimensional scaffolds made from natural biomaterials, researchers aim to recreate the texture and structure of animal muscle tissue. However, this process is still in its infancy, and the resulting disorganized muscle fibers fall short of replicating the complexity of real meat. Key challenges include replicating blood vessels, nerve tissue, connective tissue, and intramuscular fat, which influence the flavor, texture, and overall quality of meat. Currently, the most effective growth medium for cultured meat production relies on fetal bovine serum (FBS), a substance derived from the blood of deceased calves [24]. While FBS is rich in essential nutrients, hormones, and growth factors required for cell proliferation, its use is ethically controversial and economically unsustainable. Researchers are actively working to develop plant-based or synthetic alternatives to FBS that are both cost-effective and capable of supporting efficient cell growth [25]. The search for an alternative medium is vital to scaling up production and making cultured meat accessible to a wider consumer base. Health and safety concerns surrounding cultured meat also require careful consideration [26]. Advocates argue that cultured meat is inherently safer than conventional meat, as it is produced in sterile, controlled environments that eliminate exposure to intestinal pathogens like *E. coli*, *Salmonella*, and *Campylobacter* [27]. Despite these advantages, the use of animal-derived serum introduces potential risks, such as contamination with prion proteins, which are associated with transmissible spongiform encephalopathies, including Creutzfeldt-Jakob disease [28]. Additionally, the extensive cell multiplication involved in production raises concerns about unintended biological changes, such as dysregulation of cell lines, which could have unknown effects on human health [29]. Rigorous safety testing and regulatory oversight will be essential to address these uncertainties, particularly as the technology scales up for mass production. Cultured meat offers unique opportunities to enhance the nutritional profile of meat products. By modifying the composition of the growth medium, producers can control the types and ratios of fats in the final product [30]. For instance, saturated fats can be replaced with healthier polyunsaturated fats, such as omega-3 fatty acids. However, this modification introduces the challenge of managing the increased risk of rancidity associated with polyunsaturated fats. Furthermore, the bioavailability of micronutrients like iron and the organization of biological compounds within cultured cells may differ from those in conventional meat, potentially reducing their health benefits [31]. Research is needed to optimise the nutritional composition of cultured meat and ensure it provides comparable, if not superior, benefits to traditional meat. Another critical consideration is the potential for allergic reactions. Cultured meat contains the same molecular components as traditional meat, making it likely to trigger allergies in individuals sensitive to specific types of meat [32]. The major concerns raised for food allergies originate from sensitization to specific carbohydrate structures (e.g., alpha-galactose) [33]. For example, people with alpha-gal syndrome, a rare allergy to a sugar molecule found in red meat [34], or those allergic to poultry, may experience adverse reactions to cultured products derived from these sources [35]. Symptoms of such allergies range from mild skin irritations to severe anaphylaxis, highlighting the importance of clear and accurate labelling. Regulatory definitions of cultured meat, which may differ across regions, will play a crucial role in ensuring consumer safety and avoiding confusion. Notably, some meat industry stakeholders oppose the use of the term “meat” for cultured products, adding to the complexity of labelling and marketing these items. As research and development continue, cultured meat holds promise for addressing some of the most pressing challenges of modern food systems, including sustainability, animal welfare, and global food security. However, significant technical, regulatory, and societal hurdles remain. The transition from experimental production to large-scale commercialization will require breakthroughs in cell culture technology, cost reduction, and public acceptance. Equally important will be the establishment of robust safety standards and transparent labelling practices to build consumer trust and confidence. With continued innovation and collaboration across scientific, industrial, and regulatory domains, cultured meat could redefine the future of food.

### 3.2. Seaweed

Seaweed is one such alternative source of protein that requires greater investigation, and it is a promising biomass with applications in food, beverages, feed, nutraceuticals, and pharmaceuticals, also suggested as an alternative solution for the food industry to replace meat as a source of gelatine and protein [36]. Seaweed consists of several macroscopic and microscopic species of algae that can be classified based on their pigmentation into brown (Phaeophyta), green (Chlorophyta), and red (Rhodophyta) algae. It provides various amounts of protein, ranging from 3 ± 15% dry weight for brown alga, 9 ± 26% dry weight for green alga, and up to 47% dry weight for red alga [37]. Edible seaweeds are very popular in East Asia, particularly in Korea, China, and Japan. Some of the most common species used in food preparation include the red seaweeds Porphyra (Nori) for sushi wrappings, Palmaria Palmata (Dulse), and Chondrus Crispus (Irish moss) as a thickening agent. Varieties of brown seaweeds, such as the kelps Undaria Pinnatifida (Wakame) and Laminaria Japonica (Kombu), and green seaweeds, such as Ulva spp (sea lettuce), are consumed as sea vegetables [38]. Seaweed can incorporate essential amino acids, fibres, vitamins, carotenoids, and other bioactive metabolites in foodstuffs, besides acting as flavouring agents with umami taste and binding agents for partial replacement of egg yolk in formulations of vegetarian hamburgers and other products [39,40,41,42]. Within this context, edible seaweed has been considered a “superfood”. The microalgae A. platensis and C. vulgaris currently represent the largest share of the global market for algae-based proteins. Species of the genera Chlorella and Arthrospirahave been sold as ingredients in various forms, including powder, paste, pellets, and flakes, for the manufacture of pasta, beverages, vegetarian meat analogues, and animal feed. Seaweed farming is primarily conducted in Asia (China, Indonesia, the Philippines, Korea, and Japan) and Africa (Zanzibar and Madagascar), where it presents a socio-economic opportunity for coastal communities. There are significant differences in the composition of these new protein sources: Seaweed, along with insects, stands out by their superior content of proteins (50%) and fibres (27%), while they have a low content of lipids (8%); carbohydrates are present in concentrations below 20% [3]. Polyunsaturated fatty acids, especially ω-3, and phytosterols are usual seaweed constituents and corroborate several functional food claims. As well as plants, these organisms are unable to synthesize cholesterol, a substance present in foods of animal origin, like red meat and eggs. Thanks to its composition, seaweed can exert antimicrobial, antioxidant, anti-hypertensive, antidiabetic, antidepressant, antitumor, and immunomodulatory activities [43]. However, there are some doubts about the possibility of harmful effects on human health resulting from the consumption of seaweeds. Factors like the seaweed type, cultivation and harvest conditions, and further processing, among others, were found to influence the presence of hazards in seaweed. Recent literature shows that (inorganic) arsenic, cadmium, lead, mercury, iodine, pathogenic bacteria like *Salmonella* spp., *Bacillus* spp., *Vibrio* spp., and norovirus have been reported as relevant hazards in seaweed. Despite the wealth of research on these hazards, data gaps on the presence of other hazards exist. Banach, Hoek-van den Hil, and van der Fels-Klerx (2020) indicated data gaps for pesticide residues, dioxins and polychlorinated biphenyls, brominated flame retardants, PAHs, pharmaceuticals, marine biotoxins, allergens, micro-and nanoplastics, and pathogenic microorganisms *E. coli*, *Listeria* spp., *St. aureus*, norovirus, and hepatitis E virus. Moreover, although research describing the effects of processing, like washing seaweed to reduce microplastics or marine biotoxins, as well as soaking or washing and cooking to reduce inorganic arsenic and iodine in seaweed, has been reported, there remain research gaps on the effects of processing. Also, additional monitoring of seaweed and data collection on the health effects, e.g., dietary exposure, to further characterize hazards is needed [44,45,46]. With the expected further increase in protein alternatives in consumers’ diets, the risk of food allergens is apparent. The consumption of new proteins or the increased introduction of existing ones, through the use of purified protein concentrates, can lead to: (a) increased prevalence of known allergies through the extraction of proteins from their natural matrix and the incorporation of them into other products in much higher quantities; (b) development of new allergies, as a result of consuming foods that we do not consume or rarely consume, through primary sensitization to these foods or cross-reactivity reactions to existing allergens; (c) sensitization to new proteins with cross-reactivity phenomena toward foods considered non-allergenic or rarely allergenic [43]. Seaweed allergy has been reported on several occasions: cases of respiratory reactions have been described where algae acted as inhalant allergens both in children and in adult workers in a British factory producing alginates [47,48]. A case of fatal anaphylaxis possibly due to an allergy to an alginate-based paste used for dental impressions has been described in a patient with multiple severe comorbidities [49]. Anaphylaxis to antacid tablets [50] and alginate dressing [51] have also been reported (Sodium alginate is derived from brown seaweed). There are also multiple reports of hypersensitivity reactions after laminaria stick insertion (derived from the brown seaweeds Laminaria japonica or digitata) during dilation and evacuation procedures in obstetrics, which are suggestive of an underlying IgE-mediated mechanism [52]. A case of food allergy to seaweed was described in 2018 by I. Thomas and colleagues [36]. It involved a 27-year-old patient who had eaten sushi on several occasions and experienced repeated generalized reactions. The patient ingested the various ingredients of the sushi after the reactions without any issues, except for crustaceans, sesame, and seaweed. The Prick-by-Prick test with fresh tiger prawn and sesame was negative. The Prick-by-Prick test with fresh Nori (red seaweed) was positive, with a 13-mm wheal. The Prick-by-Prick test with other red seaweeds was positive, while it was negative with brown and green seaweeds, indicating the lack of cross-reactivity between different groups of seaweeds. However, patients allergic to seaweed should avoid all products containing seaweed unless tolerance has been proven with negative skin and challenge tests. Phycobiliproteins, the main proteins of red seaweeds, and phycolectins are possible allergens [53]. In a recent literature review, Gromek et al. describe five cases of allergy to spirulina (Arthrospira platensis) which is a blue-green microalgae (cyanobacteria), used as a human protein supplement because of the high content of proteins, γ-linoleic acid and vitamins. The discovered descriptions of cases were in the form of case reports, letters, and brief communications or poster presentations. All the cases were reported in Europe (France, The Netherlands, Switzerland, and the UK) An allergy to spirulina was reported, for the first time, in a 14-year-old male who developed urticaria, labial edema, and an asthma attack six hours after ingesting five spirulina tablets, A possible allergen was identified using Western blotting in the β-chain of C-phycocyanin (a cyanobacterial pigment) The second patient was identified in the Netherlands: a 17-year-old male who developed a systemic reaction 10 min after ingesting a 300 mg spirulina tablet. Another report from the UK documented two patients who had an allergic reaction to ingested spirulina powder in a smoothie. The last and most recent case of an allergy to spirulina was reported in Switzerland in a 48-year-old woman who experienced an episode of allergy to spirulina 3 and 7 h after ingesting three tablets of spirulina (400 mg each). Diagnosis was made based on the clinical symptoms of allergy, a positive SPT, OC, and basophil activation test. According to the WAO classification from 2020, the symptoms of the first four patients can be identified as anaphylaxis to spirulina. The six proteins identified in a 2022 study by Bianco et al. using proteomic and in silico analyses following WHO/FAO guidelines shows significant homology with proteins already categorized as allergens in other allergen sources. At the moment there is a notable deficiency in epidemiological data regarding hypersensitivity to spirulina and algal species more broadly [54].

### 3.3. Single-Cell Proteins

Proteins play a fundamental role in the human diet and, compared to other nutrients, are the most expensive [55,56]. The rising demand for proteins caused by population growth and the lack of other cultivable land has driven research into alternative sources to meat, dairy, and plants [57,58,59,60,61]. Single-cell proteins (SCP) represent a possible solution to meet global food needs sustainably [56]. They include proteins of high biological value with all essential amino acids, fibers, minerals, and vitamins, and are low in fats, making them a promising option for both human and animal nutrition [56,57,58,59,60,61,62,63,64]. SCP are derived from microorganisms, most frequently yeasts (Saccharomyces cerevisiae and Kluyveromyces marxianus), but also from fungi, algae, and bacteria [56,57,58]. The advantages of using microorganisms to produce SCP are many. Firstly, their rapid proliferation ensures large yields of protein. They require small spaces and minimal water, and their production is unaffected by weather conditions or soil contaminants such as insecticides or fertilizers [57,58]. SCP can be produced from a wide range of microorganisms, but as they are intended for human consumption, their selection depends on various factors, including taste and pathogenicity [57,58,61]. Some microorganisms can produce toxins or contain nucleic acids (RNA) or heavy metals, which pose risks to human health and may cause nephrotoxicity, immunosuppression, carcinogenicity, and teratogenic effects [42,60,61]. For example, high levels of RNA can lead to hyperuricemia, and many Fusarium species produce mycotoxins harmful to the central nervous system [61]. Acute tubulointerstitial nephritis has also been reported in a boy who consumed Chlorella tablets for three months [61]. The potential for allergic reactions remains a concern, especially due to possible cross-reactivity with fungal allergens and mould proteins [65]. Moreover, certain fungal proteins—such as enolase and triose-phosphate isomerase—are recognised as cross-reactive allergens. This suggests that individuals already sensitised to airborne fungi or other fungal allergens could react negatively when consuming mycoprotein [66]. Allergic reactions, including anaphylaxis, have been documented, such as an incident involving an adolescent after taking Spirulina tablets containing phycocyanin [60,61]. The choice of microorganisms is crucial, as the processes used to remove toxins or contaminants can increase costs and reduce yields [59]. Advances in production techniques have led to second-generation SCP derived from food waste, by-products, and genetically modified microorganisms [59,62]. These sources are ideal substrates as they reduce industrial residues and manufacturing costs [59,60,61,62]. SCP is not a novel concept. Initially, it was produced during World War I in Germany as yeast biomass. The bacterial SCP from methanol, Pruteen, was developed in the 1970s, and Pekilo^®^, a mycoprotein from pulp and paper industry by-products, was available in Finland from 1974 to 1989 and is now being refined [67]. Today, various SCP products are available, including Quorn™, a fungus-based meat substitute, FeedKind^®^, Spirulina, and UniProtein^®^ [67]. Additionally, KnipBio Meal, a fish substitute, was approved by the Food and Drug Administration in 2021. It consists of an SCP produced by the yeast Yarrowia lipolytica, genetically modified to enhance its nutritional profile with increased astaxanthin [61]. The potential of SCP as a sustainable and efficient protein source is substantial and offers promise for the future of food and feed industries, despite remaining challenges such as consumer acceptance, scaling production to meet global demand, and improving cost-efficiency in large-scale SCP manufacturing [59].

### 3.4. Plant-Based Proteins

Recently, the eating habits of Western countries have been changing, influenced by new trends and needs, leading to a widespread adoption of plant-based diets. On one side, vegetarianism, veganism, and their variants are gaining greater acceptance among the general population and are often chosen by adults and introduced to children even at early ages. On the other hand, emerging trends advocate plant-based alternatives for improved eco-sustainability and better health profiles within the population’s lifestyles [65]. This has driven food industries to develop new options that mimic the texture, mouthfeel, and satiety usually associated with animal-derived products, expanding choices from legumes to nuts, seeds, cereals, and tubers [66]—Table 1. However, vegetable protein sources used in these alternatives are responsible for a significant proportion of food allergies. A potential risk for consumers is the unintentional introduction of allergens through new exposures, such as novel or modified proteins, undeclared added allergens, or cross-reactivity [67]. Technologies now process vegetables like peas, potatoes, seaweed, fava beans, and chickpeas to produce isolated and concentrated protein sources that can be added to various foods to create high-protein plant-based products, reducing reliance on traditional sources like wheat and soy [68]. Potatoes, for example, are generally considered allergen-free [69]. Conversely, legumes exhibit a high degree of cross-reactivity among each other in individuals allergic to one, although it is rare for a person to be allergic to all of them [70,71]. Consequently, legume crops have expanded recently to meet market demand, despite the well-documented allergenic potential and cross-reactivity of these proteins, especially in adults, who are the primary sufferers of food allergies due to the more recognisable nature of plant-based proteins compared to animal proteins from the immune system’s perspective [72,73]. Further clinical research is needed to clarify the allergenic roles of legumes such as lentils and chickpeas, and seeds like poppy and pumpkin, particularly given their increasing popularity in diets alongside reports of allergic reactions [74]. It is important to note that the food industry often adds plant-based ingredients during product formulation to offer healthier alternatives (e.g., pea protein versus lupine or soy) or to reduce costs. When present in amounts below 2%, these ingredients are not required to be listed unless they are major allergens. This increases the risk of hidden or undeclared allergens [75,76,77]. The food industry should invest in proteomic technologies to accurately identify protein contents and improve quality control to ensure consumer safety [78,79]. Sensitisation to proteins can occur via different routes: oral, respiratory, and skin exposures, each playing a distinct role in either triggering reactivity or promoting immune tolerance. For instance, recent evidence suggests that respiratory sensitisation through aeroallergens often precedes the development of food allergies due to cross-reactivity between shared allergen components in plants. Globalisation has further influenced dietary habits worldwide, integrating foods from diverse cultures and increasing exposure to new allergens and allergy prevalence [80]. Additionally, the allergenic potential depends on the protein’s characteristics, such as secondary and tertiary structures (antigenic determinants), enzymatic activity, post-translational modifications (like glycosylation), and stability to digestion or heat. The matrix in which allergens are embedded also affects their reactivity, as does food processing [81]. For example, non-thermal techniques used for whey protein (WP) can reduce its allergenicity [82], while thermal processing involving the Maillard reaction can increase the immunogenicity of wheat proteins [83,84]. Current literature lacks comprehensive information on how various technological and cooking methods, or the composition of processed foods, influence the allergenicity of plant-based proteins. These proteins are often processed with novel techniques to create new shapes, textures, flavors, and effects for commercial purposes [85,86]. New processing methods may pose allergenic risks, as their effects on the immune system are not yet fully understood or predictable. An example is the recent rise in peanut allergies among children in Asia, potentially linked to exposure to differently processed peanuts (such as roasted peanut butter instead of boiled peanuts) [87].

#### Plant-Derived Food Allergen Components

Plant-derived food allergens mainly fall into four protein families or superfamilies: prolamin, cupin, Bet v 1/pathogenesis-related protein 10 (PR-10), and profilin (Table 1) [88]. The prolamin superfamily includes 2S albumin, non-specific lipid transfer protein (ns-LTP), cereal prolamin, and α-amylase/trypsin inhibitors. The cupin superfamily encompasses proteins such as vicilin (7S globulin) and legumin (11S globulin), found in microorganisms, plants, and animals, all originating from a common ancestral, conserved protein sequence. Bet v 1/PR-10 and profilin are major fruit allergens responsible for cross-reactive pollen allergies. Oleosins are stored in plant seeds [89]. Plant defensins, like those in the PR-12 family, contribute to innate immune responses through antifungal and antibacterial activities, and have been identified as peanut allergens [90,91]. 2S albumin, acting as a storage protein, is an allergen present in peanuts, soybeans, nuts, sesame, and buckwheat. The sequence similarity of 2S albumin across peanuts, nuts, and sesame seeds is relatively low, around 20–60%, whereas in walnut (Jug r 1) and pecan (Car i 1), both from the Juglandaceae family, the homology is higher, approximately 88% [92]. Ara h 2, Ara h 6, and Ara h 7 show limited sequence overlap, not exceeding 55%, despite being isoforms [93]. The storage proteins called legumins are found in the seeds of various plants, sharing high sequence identities ranging from 32% to 95% [94,95]. Like 2S albumin, vicilins are storage proteins. Oleosins are stored in lipid droplets within seeds, providing energy during plant growth. Oleosins act as allergens in peanuts (Ara h 10, Ara h 11, Ara h 14, Ara h 15), sesame (Sesi 4, Sesi 5), and hazelnuts (Cor a 12, Cor a 13, Cor a 15). Peanut defensins (Ara h 12 and Ara h 13), soybean defensin (Gly m 2), share low sequence identity with a reduced risk of cross-reactivity [96]. Regarding wheat allergens, cereal prolamin and alpha-amylase/trypsin inhibitors merit mention. Wheat prolamins include gliadin and glutenin, which are key components of wheat gluten [96]. The primary allergenic risk for children involves wheat components such as gliadin (Tri a 20 and 21) and low-molecular-weight glutenin (Tri a 36). Additionally, gliadin (Tri a 19) and high-molecular-weight glutenin (Tri a 26) are often implicated in food-dependent exercise-induced anaphylaxis episodes [97]. These molecules feature repetitive sequences rich in proline and glutamine, which seem to be involved in allergic symptoms. Wheat alpha-amylase/trypsin inhibitors are responsible for baker’s asthma episodes. Several types have been classified based on their assembly, and their immunoreactivity varies: monomeric (Tri a 15), dimeric (Tri a 28), and tetrameric (Tri a 29, Tri a 30, Tri a 40) [98]. Cross-reactivity with Bet v 1/PR-10 is known to contribute to plant-based allergy symptoms, as many fruits and vegetables cross-react with pollen, a condition known as pollen-food allergy syndrome (PFAS) [99]. Bet v 1 of birch pollen acts as the primary antigen, inducing allergic responses through cross-reactivity. PR-10 proteins are found in seeds, especially peanuts and soybeans. An allergy to soybean milk often coincides with elevated IgE antibodies specific to Gly m 4, leading to soybean allergy in adults [100]. Bet v 2, the first profilin identified in birch pollen, shares highly conserved amino acid sequences with other seed profilins, which are responsible for PFAS [101]. Ns-LTP, a defense protein in the PR-14 family, is detected in seeds and fruits of Mediterranean Europe, including Rosaceae fruits, vegetables, walnuts, hazelnuts, and peanuts. It is often linked to peach and apple allergies and associated with systemic symptoms [102]. GRPs, members of the snakin/GASA family, are plant antimicrobial peptides [103]. These stable proteins withstand heating and digestion. Peach GRP (Pru p7) is notably associated with systemic symptoms in peach allergy patients from France and other countries [104,105,106]. Other fruit allergies also involve cross-reactivity with homologous GRP epitopes, especially following initial sensitisation to GRP in cedar pollen [107].

### 3.5. Insects

The rapid growth of the human population and the rising demand for high-quality protein sources have prompted the scientific community to investigate novel, sustainable, and natural alternatives, including invertebrates such as insects. Currently, insects are a regular part of the diet for about 2 billion people worldwide, with over 2000 edible species identified [108]. Among these, the most commonly utilised for their protein content belong to the orders Coleoptera (beetles), Lepidoptera (caterpillars), and Hymenoptera (wasps, bees, and ants). Insects are recognised as a rich source of essential nutrients, including polyunsaturated fatty acids, essential amino acids, micronutrients, and proteins [109]. Their nutritional profile features a high protein content, along with significant amounts of fats, minerals, vitamins, and energy. A comprehensive review by Rumpold et al. highlights both the nutritional benefits and potential risks associated with edible insects [110]. Proteins are the predominant macronutrient, with an average content of approximately 40%, though this can vary from 20% to over 70%, depending on the species [110]. Notably, three insect species commonly farmed in Europe—Tenebrio Molitor, Gryllodes Sigillatus, and Schistocerca gregaria—exhibit particularly high protein levels, estimated at 52%, 70%, and 76%, respectively [108]. For instance, protein extracts derived from Hermetia Illucens and related species show poor solubility in aqueous media, which may limit their nutritional and functional potential [111]. Conversely, proteins from Protaetiabra Revitarsis and Allomyrina Dichotoma display superior thermal stability and emulsification capacity compared to T. Molitor, making them more suitable for food processing applications [112]. Additionally, the high polyphenol content in Hermetia illucens protein extracts has been shown to reduce emulsification activity, further influencing their functional properties [113]. Beyond their protein content, insect-derived proteins exhibit variable digestibility. They are generally more digestible than plant proteins but slightly less so than traditional animal proteins [114]. Moreover, their functional properties differ significantly between species. The use of insect ingredients, such as insect flour, has been explored in food formulations, particularly in bakery products, where it enhances both nutritional value and overall product quality [115]. An equally important aspect of insect consumption is its sustainability. According to studies by Mason et al., producing one gram of beef requires 21 times more water than producing the same amount of protein from crickets, making insect-derived proteins a valuable and sustainable alternative to conventional meat sources [116]. However, consuming insects carries potential risks. Concerns include chemical hazards, such as heavy metal accumulation, and microbiological contamination. Additionally, insects have been linked to allergic reactions through various exposure routes, including inhalation, direct contact, stings or bites, and ingestion [117]. Data on allergic risks linked to insect consumption remain limited. Most studies involve small sample sizes (*n* < 20), and geographic disparities in research availability hinder comprehensive evaluation [118]. Many relevant articles are published in Chinese, limiting access for Western researchers, while data from regions where insect consumption is common—such as Africa, South America, and Asia—are scarce [119]. Few studies have explored the epidemiology of food allergies to insects, despite their inclusion in many populations’ diets worldwide. In China, a review of literature from 1980 to 2007 estimated that around 17% of food-related anaphylaxis cases were due to insect consumption, with locusts, grasshoppers, and silkworms being the most frequently implicated species [120]. It has been reported that over 1000 patients annually suffer anaphylactic reactions after consuming silkworm pupae in China [121]. Conversely, a 2015 study in Laos found that 7.6% of insect consumers had experienced an allergic reaction [122]. The remaining literature mainly comprises case reports, with about 30 documented cases [119]. To date, many IgE-binding allergens have been identified in the main groups of invertebrates, including mites, crustaceans, arachnids, insects, mollusks, and nematodes. Most of these are pan-allergens common across invertebrate groups, as shown in Table 1 [119]. The potential allergenicity of insect-derived proteins is a serious concern, especially for individuals sensitised to invertebrate allergens such as those found in crustaceans and house dust mites (HDM) [123]. The most clinically relevant insect allergens are tropomyosin (TM) and arginine kinase (AK), both of which have demonstrated cross-reactivity with other invertebrate allergens [124]. TM is the most widely studied insect allergen, characterised by an alpha-helical structure and a key role in muscular contraction [125]. As a recognised allergen, TM is commonly found in shellfish, including crustaceans, mollusks, and cephalopods [126]. For example, a study by Jeong et al. showed that silkworm tropomyosin shared 73.5 to 92.3% sequence identity with other allergenic tropomyosins, suggesting possible cross-reactivity [127]. Furthermore, Kamemura et al. identified a high molecular weight cricket tropomyosin that appears cross-reactive with shrimp tropomyosins, highlighting the strong amino acid similarity among insect homologues [128]. Recently, TM from the yellow mealworm (YMW) (Tenebrio Molitor) has been identified as a major allergen in this emerging food source [129]. The cosensitisation between YMW and crustaceans has been extensively demonstrated, particularly in individuals allergic to shrimp [130]. Barre et al. conducted a comprehensive study on YMW allergenic proteins, identifying TM, AK, α-amylase, heat shock protein 70 (HSP70), apolipophorin-III (apoLp-III), larval cuticle protein (LCP), and a 12-kDa hemolymph protein as allergenic targets for shrimp-allergic patients [131,132]. Furthermore, most shrimp-allergic individuals were found to be sensitised to T. Molitor [133]. Additional research by Broekman et al. identified three novel mealworm allergens: larval cuticle proteins A1A, A2B, and A3A [130]. Notably, the allergens causing primary mealworm allergy may differ from those responsible for cross-reactivity [126]. AK is another key pan-allergen, with enzymatic activity and a highly conserved amino acid sequence across invertebrates [127]. Structurally, AK consists of a β-sheet domain surrounded by α-helices. For example, sensitisation to silkworm allergens is mainly linked to AK (Bomb m1), paramyosin, and chitin. AK Bomb m1 is the only officially recognised edible insect allergen listed in the IUIS allergen database (www.allergen.org accessed 16 March 2025) [134]. Cricket allergens remain poorly characterised, with limited research on cross-reactivity in those allergic to shellfish or HDM. Srinroch et al. identified AK and hexamerin 1B (HEX1B) as major and minor allergens in Gryllus bBimaculatus, using sera from prawn-allergic patients [135]. Beyond TM and AK, additional allergens have been identified in Bombyx mori, including chitinase and paramyosin [136,137]. Karnaneedi et al. found further potential allergens, demonstrating cross-reactivity among hemocyanin, vitellogenin, HSP20, apolipophorin-III, and chitin-binding proteins present in shrimp, crickets, and black soldier fly. Troponin C, involved in muscle contraction, has also been implicated as a relevant pan-allergen in insects such as Bombyx and Drosophila [138]. Numerous reports describe sensitization to cockroaches, though these studies mainly focus on inhalant exposure rather than ingestion. No cases of allergic reactions following their consumption have been documented [123]. Beyond allergenicity, other health risks linked to insect consumption include the presence of heat-resistant thiaminase in pupae of the African silkworm (*Anaphe* spp.), which has been associated with seasonal ataxic syndrome due to thiamine deficiency in Nigeria [139]. The impact of food processing on allergenic potential is another concern, as it can alter protein structure and IgE-binding capacity. However, limited studies have assessed the effects of processing on edible insects [140]. Some research has examined thermal treatments on mealworms, silkworms, and locusts, while enzymatic hydrolysis has been investigated as a method to reduce allergenicity in food matrices [140,141,142]. In the case of mealworms, processing methods affected protein solubility but did not consistently decrease immunoreactivity.

### 3.6. Regulation of Alternative Protein Consumption

The main European food regulations impacting the development and commercialization of alternative proteins include the Novel Food Regulation, the regulation of genetically modified (GM) foods, food information regulations, nutrition regulations, and the Organic Food Regulation [142,143,144]. In the European context, according to Regulation (EU) 2015/2283, novel foods are defined as food products and substances that had no ‘significant’ history of consumption in the EU before 15 May 1997 [145]. These products must undergo an authorization process to assess their safety before being placed on the market. EFSA is responsible for pre-market risk assessment. To introduce a novel food to the market, a dossier must be submitted containing specific studies on the food’s effects on human health. This process follows a centralized procedure. EFSA’s scientific opinion usually serves as the basis for the European Commission to draft and adopt food legislation applicable across all member states. In the European Community (EU), the insects are considered “novel foods” [145]. Regulation (EU) 2015/2283, effective from 1 January 2018, replaced Regulations (EC) No 258/97 and No 1852/2001, and streamlined the approval process for innovative and novel foods while ensuring consumer safety [145]. The EU Novel Foods Regulation, in force since 1 January 2018, simplifies the introduction of novel and innovative foods into the EU market, while ensuring safety for consumers. EFSA has approved the market placement of several insect species under Regulation (EU) 2015/2283. In Italy, four insect species were authorized in 2023 for consumption in frozen, dried, and powdered form: *Tenebrio molitor* (yellow mealworm), *Locusta migratoria* (migratory locust), *Acheta domesticus* (house cricket), and *Alphitobiusdiaperinus* (lesser mealworm) [146,147,148,149,150]. These insect-based products must carry clear labelling indicating the insect species used, the quantity, the country of origin, and potential allergy risks. Additionally, these products must be sold in designated compartments with appropriate signage. As of 2024, the EU has approved 20 new seaweeds as novel foods, increasing the total number of seaweeds authorised for consumption to 60. Cultivated meat has not yet been approved for sale in the EU, but has been authorised for consumption in Singapore since 2020, and in the US in 2023. The European Commission has clarified that mycoproteins derived from *Fusarium venenatum* can be considered traditional foods. Therefore, under specified conditions, products containing these mycoproteins do not require prior authorization under Regulation (EU) 2015/2283.

### 3.7. Allergenicity Assessment of Alternative Proteins

As the use of alternative protein sources, such as insects and legumes, expands, concerns are increasing over their allergenic potential and the risk of cross-reactivity. Many of these proteins resemble known allergens structurally, which raises the likelihood of cross-reactive allergic reactions. For example, insect proteins like tropomyosin share structural similarities with crustacean proteins, meaning individuals allergic to shrimp or crabs could experience reactions [127]. Similarly, plant-based proteins from chickpeas and lentils have epitopes in common with peanuts, posing potential risks to peanut-allergic persons (Table 2) [93]. To assess these risks, a comprehensive evaluation is carried out, which includes:(1)Gene source analysis: identifying the origin of the protein(2)Sequence homology: comparing amino acid sequences with recognised allergens(3)IgE binding tests: evaluating how proteins interact with IgE antibodies from allergic subjects(4)Pepsin resistance testing: assessing protein stability in simulated gastric fluid, as resistant proteins are more likely to be allergenic.

Despite these methods, accurately identifying allergenic determinants at the molecular level remains difficult. There is currently no single, definitive approach to fully predict the allergenicity of a new protein. Besides classical in vivo (e.g., human or animal testing) and in vitro (lab) assessments, in silico (computational) tools are becoming increasingly significant. These include:(1)Bioinformatic screening: utilised to identify potential allergenic sequences by comparing new proteins to known allergens using databases such as AllergenOnline, Allermatch, or UniProt. Criteria include at least 35% identity over an 80-amino acid segment or eight or more consecutive identical amino acids.(2)Omics technologies, especially:
○Proteomics is the large-scale study of the complete set of proteins (proteome) expressed by a cell or tissue. Techniques include:
Mass Spectrometry (MS): for identifying and quantifying proteins.2D-Gel Electrophoresis (2D-GEL): for separating proteins by charge and mass.LC-MS/MS (Liquid Chromatography with Tandem Mass Spectrometry): for in-depth protein analysis.○Transcriptomics: the investigation of RNA transcripts to comprehend gene expression.


These modern techniques enhance traditional testing methods, offering a more comprehensive and predictive approach to assessing the allergenic potential of alternative protein-based foods [151]. Thorough studies, involving both in vitro and in vivo methods, will be essential to evaluate how effective these strategies are in reducing the allergenic potential of emerging protein alternative sources.

### 3.8. Alternative Proteins Projections to 2050

A study by Witte et al. (2021) estimates that by 2035, alternative proteins—including plant-based options—will make up 11% of the global food protein market, with the potential to reach as high as 22%. The study identifies Europe and North America as the most mature markets for alternative proteins, while pointing to the Asia-Pacific region as having the greatest growth potential. By 2035, Asia-Pacific is expected to account for two-thirds of global consumption of alternative proteins [4]. According to the study’s baseline scenario, Europe could reach 15 million metric tonnes of alternative proteins by 2035. This includes plant-based meat alternatives, cultured meat (derived from animal cells), and products made using microorganisms such as microalgae. Insect proteins are not included in the analysis. This volume would represent about 22% of the market currently held by conventional animal proteins. In an optimistic scenario—where Europe reaches ‘peak meat’ consumption by 2025 and benefits from significant technological and regulatory advances—alternative proteins could grow to 33 million metric tonnes, representing 34% of the current conventional protein market. A study by Witte et al. estimates that by 2035, alternative proteins, including plant-based options, will make up about 11% of the global food protein market, with the potential to reach as high as 22%. The study highlights Europe and North America as the most mature markets for these alternatives, while pointing out that the Asia-Pacific region has the greatest growth potential. By 2035, Asia-Pacific is expected to account for two-thirds of worldwide consumption of alternative proteins [4]. According to the baseline scenario in the study, Europe could produce around 15 million metric tonnes of alternative proteins by 2035. This includes plant-based meat substitutes, cultured meat (made from animal cells), and products created using microorganisms like microalgae. Proteins from insects are not included in this analysis. This volume would represent about 22% of the current market share held by traditional animal proteins. In an optimistic scenario—where Europe reaches ‘peak meat’ consumption by 2025 and benefits from significant technological and regulatory advances—alternative proteins could grow to 33 million metric tonnes, making up roughly 34% of the current conventional protein market [152].

## 4. Conclusions

The incorporation of alternative proteins into human diets represents a promising solution to many of the challenges facing the global food system today. With growing concerns over environmental sustainability, food security, and health, plant-based proteins, cultured meat, and insect-derived proteins offer viable, lower-impact options compared to traditional animal sources. While there are still barriers to overcome, such as consumer acceptance, regulatory approval, and production scalability, the continued advancement in food technology and increased awareness of the benefits of sustainable eating are accelerating progress. Ultimately, alternative proteins have the potential to play a key role in building a more resilient, ethical, and environmentally friendly food future. It is essential, however, that these new foods are safe for the consumer and do not pose any health risks, including the risk of allergies. The identification of new protein allergens in insects, seaweed, and other alternative foods is expected to be one of the major challenges in the coming years. At present, technical and methodological limitations prevent accurate prediction of the allergenic potential of novel or modified foods. There is a lack of globally recognized, science-based standardized methods for assessing the allergenicity of alternative proteins. To address this gap, it is essential to employ advanced technologies such as proteomic and bio-informatic tools. There are still regulatory gaps, with current frameworks primary targheting traditional allergens, while novel protein allergens are often overlooked. A thorough allergen database that incorporates alternative proteins is urgently needed to enhance food safety evaluations. Available literature examined in our review, presents notable limitations—including a lack of standardized methodologies, small sample sizes, and insufficient long-term data—making it difficult to draw definitive conclusions about the safety of these novel proteins for allergic individuals. As the market for alternative proteins continues to grow, more rigorous, comprehensive, and harmonized studies will be essential to understand their allergenic risks better and to guide both regulatory frameworks and consumer choices.

## Figures and Tables

**Figure 1 nutrients-17-02448-f001:**
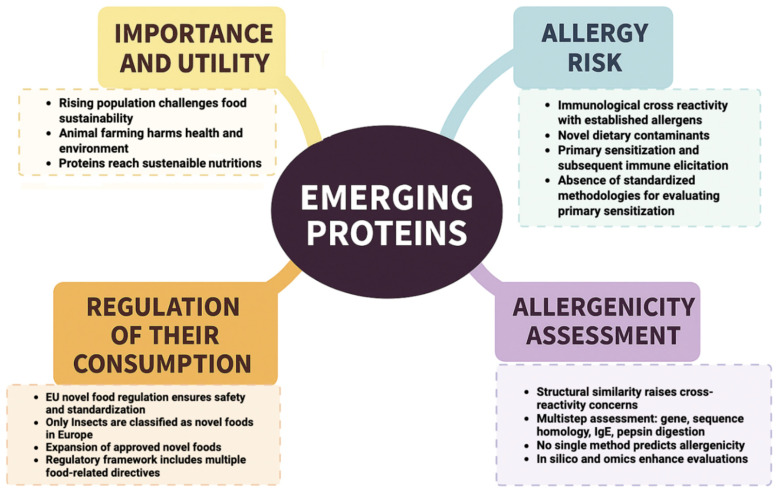
Benefits and challenges of consuming emerging proteins. Created in BioRender.com (https://www.biorender.com; access date: 9 June 2025).

**Figure 2 nutrients-17-02448-f002:**
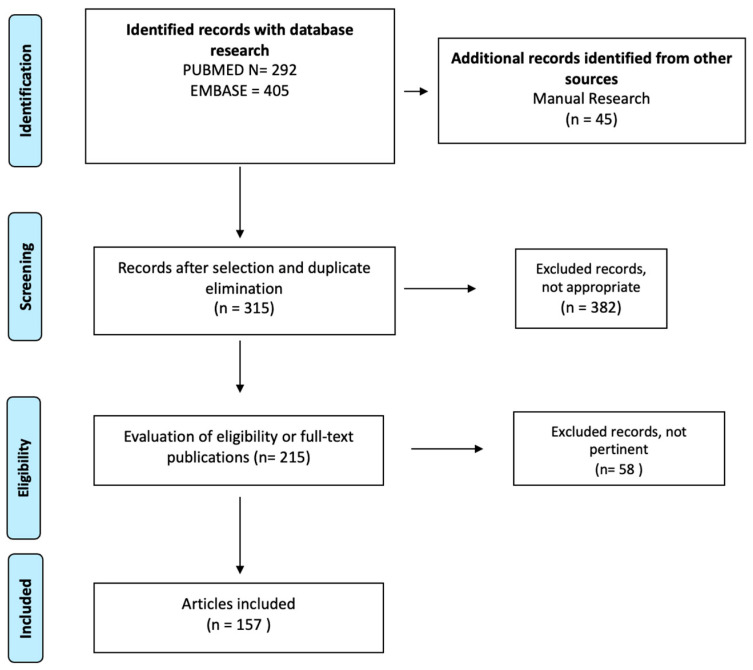
Search flow diagram.

**Table 1 nutrients-17-02448-t001:** Most used plant-derived food alternatives as sources of allergens.

Plant-Derived Foods	Tofu (Main Ingredient: Fermented Soy Milk)
	Soybeans, Tempeh (main ingredient: fermented soybeans), Edamame (immature soybeans)
	Peanuts
	Lentils
	Chickpeas
	Beans
	Peas
	Quinoa
	Mycoprotein
	Broccoli
	Tree nuts (especially almonds)
	Spirulina
Plant-derived food allergen components	Prolamin superfamily, including 2S albumin, non-specific lipid transfer protein (ns-LTP), cereal prolamin, and a-amylase/trypsin inhibitors
	Cupin, including vicilin (7S globulin) and legumin (11S globulin)
	Bet v 1/pathogenesis related protein 10 (PR-10)
	Profilin
	Oleosins

**Table 2 nutrients-17-02448-t002:** Allergenicity of protein sources.

Protein Source	De Novo Allergenicity	Cross-Reactivity Risk	Estimated Allergenicity Level	Notes
Insects	Moderate–High (e.g., mealworms, crickets)	High (shellfish, dust mites—tropomyosin, arginine kinase)	High	Strong IgE cross-reactivity with crustaceans; de novo sensitization possible with regular exposure
Single-Cell Proteins	Low–Moderate (yeast, bacteria, fungi)	Moderate (fungal/mold protein sensitization)	Moderate	True allergies are rare, but may trigger responses in mold/fungal-sensitive individuals
Seaweed	Very low (macroalgae)	Very low (microalgae: possible fungal/microbial cross-reactivity)	Low	Reactions often due to iodine or additives (e.g., carrageenan); not true IgE allergies
Cultured Meat	Low (if species-specific sensitization exists)	Variable (depends on source species: bovine, porcine, avian)	Low–Moderate	Similar allergenic profile to conventional meat if derived from same species
Plant-Based Proteins	High (especially soy, wheat, pea, lupin)	High (legume cross-reactivity, e.g., soy–pea–peanut–lupin)	High	Common allergens; frequent in children and adults; cross-reactivity within legumes is well known

## Data Availability

The authors declare that the data supporting the findings of this study are available within the paper.

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
