# Peer review of "Sustainable Nutrition and Food Allergy: A State-of-the-Art Review"

_nutrients, 2025, doi:10.3390/nu17152448_

Round 1

Reviewer 1 Report

Comments and Suggestions for Authors

The article addresses the topical and important issue of the impact of alternative protein sources on the risk of developing food allergies, focussing on five main categories: Plant proteins, in vitro meat, unicellular proteins, edible insects and algae. The article has educational potential and may be valuable for specialists in nutrition, allergology and food technology.
Advantages of the article:
Timeliness and broad coverage of the topic:
The authors precisely identify the links between global food trends, sustainable development and the growing number of food allergies.
Multidisciplinarity:
The text combines biological, technological, toxicological, immunological and regulatory aspects, making it potentially useful for a wide audience.
Rich source base:
The authors have conducted a comprehensive literature search in two databases (PubMed and Embase), covering 156 articles.
Diversity of protein sources:
Discussion of different groups of alternative proteins (including insects and algae) in the context of their potential allergenicity is rare and valuable.
Substantive caveats:
Lack of a clear methodology for risk assessment:
Although the paper is a review of the literature, no formal research quality assessment tools (e.g. PRISMA, GRADE) were used. There is also no systematic assessment of the level of evidence for individual theses.
Narrative nature of the text and lack of synthetic summaries:
The article contains many detailed case descriptions and technical data, but no clear comparative tables (e.g. on the degree of allergenicity of individual protein sources). Partial summaries would increase its usefulness.
Lack of accurate epidemiological data:
In many cases, authors refer to the occurrence of allergic reactions without providing specific frequency indicators or case numbers, making it difficult to assess the extent of the phenomenon.
Excessive reliance on case reports:
In many places, the main conclusions are based on descriptions of individual cases (e.g. allergies to spirulina or alginate), which may lead to an over-interpretation of the risk.
Linguistic and editorial inconsistencies:
The article needs revision, removal of typos and standardised formatting of citations.
Suggestions for improvement:
Include a formal assessment of allergy risk using tools such as GRADE or at least a classification of strength of evidence.
Add a summary table comparing allergenicity (de novo, cross-reactivity) for the five main protein categories.
Use a methodological framework (e.g. PRISMA) to increase the transparency of the literature selection process.
Clarify the conclusions – they are currently too general and do not provide clear guidance for clinical practise or regulatory authorities.
Add data on the incidence of allergic reactions in population studies (if available).

Author Response

Thank you very much for taking the time to review this manuscript. Please find the detailed responses below and the corresponding revisions highlighted in the resubmitted files:

Suggestions for improvement:

Point 1: Include a formal assessment of allergy risk using tools as Grade or at least a classification of strength of evidence

Response 1: Our intention was not to conduct a systematic review, but rather a narrative review, as stated in the introduction. We chose this approach because, to the best of our knowledge and given the recent interest in the topic, there are currently few studies available, most of which are case reports. However, we have now revised our manuscript to present it as a 'state-of-the-art review,' which is also reflected in the updated title. Accordingly, we have included a flowchart detailing the number of manuscripts at each step of the selection and exclusion process.

Point 2: Add a summary table comparing allergenicity (de novo, cross-reactivity) for the five main protein categories.

Response 2: We have added a summary table comparing allergenicity (de novo, cross-reactivity) for the five main protein categories (lines 675-676).

Point 3: Use a methodological framework (e.g. PRISMA) to increase the trasparency of the literature selection process.

Response 2: We have added a flowchart detailing the number of manuscripts at each step of the selection and exclusion process (PRISMA) (lines 159-160).

Point 4: Clarify the conclusions-they are currently too general and do not provide clear guidance for clinical practice or regulatory authorities.

Response 4:  We have revised the conclusions following the suggestions provided (lines 729-745)

Point 5: Add data on incidence of allergic reactions in population studies (if available)

Response 5: To the best of our knowledge, these data are not available.

Thank you for your comments and suggestions.

Thank you very much for taking the time to review this manuscript. Please find the detailed responses below and the corresponding revisions highlighted in the resubmitted files:

Suggestions for improvement:

Point 1: Include a formal assessment of allergy risk using tools as Grade or at least a classification of strength of evidence

Response 1: Our intention was not to conduct a systematic review, but rather a narrative review, as stated in the introduction. We chose this approach because, to the best of our knowledge and given the recent interest in the topic, there are currently few studies available, most of which are case reports. However, we have now revised our manuscript to present it as a 'state-of-the-art review,' which is also reflected in the updated title. Accordingly, we have included a flowchart detailing the number of manuscripts at each step of the selection and exclusion process.

Point 2: Add a summary table comparing allergenicity (de novo, cross-reactivity) for the five main protein categories.

Response 2: We have added a summary table comparing allergenicity (de novo, cross-reactivity) for the five main protein categories (lines 675-676).

Point 3: Use a methodological framework (e.g. PRISMA) to increase the trasparency of the literature selection process.

Response 2: We have added a flowchart detailing the number of manuscripts at each step of the selection and exclusion process (PRISMA) (lines 159-160).

Point 4: Clarify the conclusions-they are currently too general and do not provide clear guidance for clinical practice or regulatory authorities.

Response 4:  We have revised the conclusions following the suggestions provided (lines 729-745)

Point 5: Add data on incidence of allergic reactions in population studies (if available)

Response 5: To the best of our knowledge, these data are not available.

Thank you for your comments and suggestions.

Reviewer 2 Report

Comments and Suggestions for Authors

With real interest, I read the manuscript entitled “Sustainable nutrition and food allergy“ (publication ID: nutrients-3757082), written Anania and colleagues.

One needs to assess this manuscript in several dimensions. First of all, this work focuses on a very interesting and up-to-date topic. It is necessary to constantly look for renewable sources of food, especially considering that the constant growth of human population. One can select several approaches writing this type of review, and the Authors selected the most appropriate one, at least in my opinion, i.e. the approach based on the systematic literature search. Furthermore, it is important how the Authors summarize and present the extract of the collected knowledge, and finally, how the story based on that will be sold to the Readers, i.e. how the manuscript will be written and illustrated. In my opinion, this review article analyses and presents the results of the literature search very well, and subsequently very nicely discusses those.

I have four minor/facultative specific comments:

  1. If the Authors would like to call their review “systematic“, an additional flow chart presenting in more detail the number of the manuscripts during subsequent selection/exclusion steps would be required.
  2. Line 118. Is it really “For example“ or in facto r used search combinations are given? In the case of a systematic review (see above), all search phrases should be given.
  3. Furthermore, if this was to be considered a systematic review, a full list of 156 relevant articles of the final selection should be shown in the online supplement.
  4. Please, try to eliminate some typos and other small textual failures. For example, in line 272, you write “NORI“, but why using the upper case?

Author Response

RESPONSES TO REVIEWER 2 COMMENTS

 (REVIEWER 2)

Thank you very much for taking the time to review this manuscript. Please find the detailed responses below and the corresponding revisions/corrections highlighted in the re-submitted files:

Point 1: If the Authors would like to call their review “systematic”, an additional flow chart presenting in more detail the number of the manuscripts during subsequent selection/exclusion steps would be required.

Response 1: Our intention was not to conduct a systematic review, but rather a narrative review, as stated in the introduction. We chose this approach because, to the best of our knowledge and given the recent interest in the topic, there are currently few studies available, most of which are case reports. However, we have now revised our manuscript to present it as a 'state-of-the-art review,' which is also reflected in the updated title. Accordingly, we have included a flowchart detailing the number of manuscripts at each step of the selection and exclusion process. (lines 159-160).

Point 2: Line 118. Is it really “For example” or in factor used search combinations are given? In the case of a systematic review (see above), all search phrases should be given.

Response 2: We eliminated “For example” (line 147)

Point 3: Furthermore, if this was to be considered a systematic review, a full list of 156 relevant articles of the final selection should be shown in the on line supplement.

Response 3: We did not list the 157 relevant articles in an online supplement because we decided to convert our narrative review into a “state of the art review”.

Point 4: Please, try to eliminate some typos and other small textual failures. For example, in line 272, you write “NORI”, but why using the upper case?

Response 4: We have eliminated typos and small textual failures including NORI (line 306).

Thank you for your comments and suggestions.

Reviewer 3 Report

Comments and Suggestions for Authors

This is an interesting work conducted by Anania and collaborators. I suggest some revisions before the manuscript can be considered for publication in Nutrients.

The type of review must be indicated in the title.

Abstract: The study’s aims have to be clearly mentioned in the beginning, after a background statement, and not at the end. Applied methods, like the search strategy, searched databases, keywords, etc… are not mentioned, please revise it. Conclusions and future perspectives are also missing.

You should include jellyfish as food in your review.

To systematize the provided information, this review should include some tables with the cited studies.

Section 3 should be renamed as “Results and Discussion”.

Conclusions should be Section 4 and not 3.9.

The study’s limitations have to be discussed by the authors, and future perspectives are missing.

Author Response

RESPONSES TO REVIEWER 3 COMMENTS

 (REVIEWER 3)

Thank you very much for taking the time to review this manuscript. Please find the detailed responses below and the corresponding revisions highlighted in the resubmitted files:

Point 1: The type of review must be indicated in the title.

Response 1: We have revised the title and clarified the type of review: A state-of-the-art review.

Point 2: Abstract:  The study’s aims have to be clearly mentioned in the beginning, after a background statement, and not at the end. Applied methods, like the search strategy, searched database, keywords, etc… are not mentioned, please revise it. Conclusions and future perspectives are also missing.

Response 2: We modified the abstract by including search strategies, conclusions and future perspective and shifting the scope of the work as requested (lines 49-60)

Point 3: You should include jellyfish as food in your review.

Response 3: We did not include jellyfish due to the limited number of studies currently available in the literature, and because our focus was on the most commonly used foods in alternative diets to date.

Point 4: section 3 should be renamed as “Result and Discussion”.

Response 4: we have renamed section 3 as “Result and Discussion” (line 163)

Point 5: Conclusion should be Section 4 and not 3.9

Response 5: The requested change has been made (line 719).

Point 6: The study’s limitations have to be discussed by the authors, and future perspectives are missing.

Response 6: we have included the limitations of the study and future perspectives in the conclusions (lines 712-718)

Thank you for your comments and suggestions.

Round 2

Reviewer 1 Report

Comments and Suggestions for Authors

Thank you for your careful consideration of the comments contained in the review and for introducing the suggested corrections into the manuscript. I am pleased that the recommendations have been taken into account in a reliable and thoughtful way, which has significantly improved the scholarly quality and clarity of the text.

I appreciate your openness to criticism and your commitment to the editorial process – this attitude is commendable and conducive to the development of a reliable scientific discussion.